# Proteomic analyses identify ARH3 as a serine mono-ADP-ribosylhydrolase

Jeannette Abplanalp [1,2], Mario Leutert [1,2], Emilie Frugier[3], Kathrin Nowak[1,2], Roxane Feurer[1], Jiro Kato[4], Hans V.A. Kistemaker[5], Dmitri V. Filippov[5], Joel Moss[4], Amedeo Caflisch[3] & Michael O. Hottiger [1]

ADP-ribosylation is a posttranslational modification that exists in monomeric and polymeric forms. Whereas the writers (e.g. ARTD1/PARP1) and erasers (e.g. PARG, ARH3) of poly-ADP-ribosylation (PARylation) are relatively well described, the enzymes involved in mono-ADP-ribosylation (MARylation) have been less well investigated. While erasers for the MARylation of glutamate/aspartate and arginine have been identified, the respective enzymes with specificity for serine were missing. Here we report that, in vitro, ARH3 specifically binds and demodifies proteins and peptides that are MARylated. Molecular modeling and site-directed mutagenesis of ARH3 revealed that numerous residues are critical for both the mono- and the poly-ADP-ribosylhydrolase activity of ARH3. Notably, a mass spectrometric approach showed that ARH3-deficient mouse embryonic fibroblasts are characterized by a specific increase in serine-ADP-ribosylation in vivo under untreated conditions as well as following hydrogen peroxide stress. Together, our results establish ARH3 as a serine mono-ADP-ribosylhydrolase and as an important regulator of the basal and stress-induced ADP-ribosylome.

[1] Department of Molecular Mechanisms of Disease, University of Zurich, Winterthurerstrasse 190, 8057 Zurich, Switzerland. [2] Molecular Life Science PhD Program of the Life Science Zurich Graduate School, Winterthurerstrasse 190, 8057 Zurich, Switzerland. [3] Department of Biochemistry, University of Zurich, Winterthurerstrasse 190, 8057 Zurich, Switzerland. [4] Laboratory of Translational Research, National Heart, Lung and Blood Institute, NIH, Bethesda, MD 20892-1590, USA. [5] Leiden Institute of Chemistry, Department of Bio-organic Synthesis, Leiden University, Einsteinweg 55, 2333 CC Leiden, The Netherlands. Correspondence and requests for materials should be addressed to M.O.H. (email: michael.hottiger@dmmd.uzh.ch)

ADP-ribosylation is an evolutionarily conserved covalent posttranslational modification mainly catalyzed by ADP-ribosyltransferases (ARTs)[1,2]. These enzymes use nicotinamide adenine dinucleotide (NAD+) to transfer ADP-ribose (ADPr) moieties onto specific amino acid residues of target proteins[3], leading to mono-ADP-ribosylation (MARylation), or to build linear and branched chains of poly-ADP-ribose (PARylation)[2]. To date, at least 16 different enzymes are known to catalyze MARylation in mammals. Besides the cholera toxin-like ARTs (ARTCs) ARTC1, 2, and 5[4], also the majority of diphtheria toxin-like ARTs (ARTDs) and Sirtuins 4 and 6 have been shown to possess MARylation activity[5,6]. Only ARTD1/2 and Tankyrase1/2 (ARTD5/6) were described to form PAR chains[2].

In contrast to PARylation[1,7], the specific roles of MARylation are less well established. Nonetheless, an increasing number of studies suggests MARylation to be implicated in various cellular processes, including immunomodulation, endoplasmic reticulum (ER) stress, cytoskeleton rearrangement, cell metabolism, and host–pathogen interactions[8]. Studying the enzymes that catalyze (mono-ARTs) and demodify MARylation (mono-ADP-ribosyl(-acceptor) hydrolases, mono-ARHs) is most instructive for understanding the physiological role of MARylation. While the ARTCs have been shown to specifically MARylate arginine sites, the target amino acids modified by specific ARTDs are less well known. However, the following amino acid residues are known to be ADP-ribosylated by mammalian ARTDs: glutamate, aspartate, lysine, arginine, and serine (E, D, K, R, and S), with serine having been recently identified as the major nuclear ADPr acceptor site[9,10].

ADP-ribosylation of proteins is reversible. In mammals, two enzymes, poly-ADP-ribose glycohydrolase (PARG) and ARH3, are known to degrade PAR chains[11–13]. Whereas several cytosolic, mitochondrial and nuclear isoforms of PARG have been described[14], ARH3 seems to exist in only one isoform, which, however, was reported to be likewise present in the cytosol, mitochondria, and nucleus[15]. PARG has both endo- and exo-glycosidase activities[16–19], whereas ARH3 seems to exert only exoglycosidase activity[13]. Besides PAR, ARH3 hydrolyzes O-acetyl-ADP-ribose (OAADPR)[20,21], a product of the Sir2-catalyzed NAD+-dependent histone deacetylation reaction[22–25], to produce ADP-ribose in a time- and Mg2+-dependent reaction and thus ARH3 could participate in two signaling pathways[21]. ARH3 seems to be ubiquitously expressed in mouse and human tissues[12]. The 39-kDa ARH3 shares amino acid sequence similarity with both ARH1 and ARH2. Critical vicinal acidic amino acids in ARH3, identified by mutagenesis (i.e., D77 and D78), are located in a region similar to that required for activity in ARH1[12].

In vivo nuclear and cytoplasmic PARylation induced by hydrogen peroxide (H2O2) was rather degraded by ARH3 than by PARG, suggesting an important physiological function of ARH3 in the oxidative stress response[15]. ARH3-deficient (KO) mouse embryonic fibroblasts (MEFs) were more susceptible to H2O2-induced cytotoxicity than wild-type (WT) MEFs and ARH3 expression in ARH3 KO MEFs reduced their sensitivity toward H2O2[15].

The first enzyme described to hydrolyze MARylation was ARH1, which releases ADPr from arginine residues[26]. In contrast, ARH3 does not hydrolyze mono-ADP-ribose-arginine nor -cysteine, -diphthamide, or -asparagine bonds[12,13,21]. Recently, others and we identified the mammalian proteins MACROD1, MACROD2, and TARG/OARD1/C6ORF130 as novel glutamate- and aspartate-specific mono-ADP-ribosylhydrolases[27–29]. A hydrolase capable of releasing ADPr from other ADP-ribose acceptor sites (e.g., serine or lysine residues) has until very recently not been published[30].

We report here that, when comparing ARH3 and PARG activity, in addition to its PAR hydrolase activity ARH3 catalyzes demodification of MARylated proteins, as shown by in vitro demodification of ARTD8. Molecular modeling revealed amino acids in the catalytic cleft of ARH3 that proved important for both binding and hydrolysis of MARylated and PARylated peptides. Owing to our recent advancements in mass spectrometric ADP-ribosylome analyses, we are able to demonstrate the in vitro as well as in vivo relevance of ARH3's serine-mono-ARH activity by a specific increase in unique serine-ADPr sites in enriched ADP-ribosylated peptides and MEFs from ARH3-deficient animals as compared to WT controls, respectively. Finally, we provide a rich dataset of ARH3-targeted proteins and their corresponding ADP-ribosylation sites.

## Results

**ARH3 has mono-ADP-ribosyl-acceptor hydrolase activity.** When comparing the efficiency of PARG and ARH3 in PAR degradation, we observed that ARH3 removed radioactively labeled ADPr from automodified ARTD1 to an extent comparable to PARG but demodified transmodified recombinant H3 and H2B histone tails almost completely (Fig. 1a, left panel, Supplementary Fig. 1a). The quantification of the assay revealed that ARH3 removes 80% of the incorporated ADPr from the H3 tail, whereas PARG hydrolyzed only approximately 30% (Fig. 1a, right panel). To biochemically characterize the enzymatic reaction catalyzed by ARH3 and PARG, we performed concentration- and time-dependent experiments. ARH3, but not PARG, efficiently demodified ARTD1-transmodified H3 histone tails, suggesting that ARH3 not only degrades ARTD1-mediated PARylation but also might be a mono-ADP-ribosylhydrolase (Supplementary Fig. 1b, c). Since ARTD1 is known to poly-ADP-ribosylate its target proteins, we chose a radioactive in vitro MARylation/de-MARylation assay using the automodification ability of ARTD8 known to be a mono-ART[31]. Intriguingly, in the demodification step using either PARG or ARH3, the radioactive signal was substantially reduced (i.e., 80%) only in the presence of ARH3, but not by PARG, indicating that indeed ARH3 has in vitro mono-ARH activity (Fig. 1b).

**Various residues regulate ARH3's activity.** Having established that ARH3 has mono-ARH activity, we set out to study which amino acids of the catalytic cleft of ARH3 are involved in MAR-binding and/or MAR hydrolase activity. To identify the crucial amino acids, we performed an automatic overlap of the three-dimensional structures of ARH3 and the structurally related bacterial mono-glycohydrolase DraG (PDB code 2WOE) (Fig. 1c, left panel). The structural similarity was striking with an average deviation of only 1.4 Å for >200 pairs of corresponding Cα atoms (located mainly in the α-helices and close to the binding site of ADPr) despite the sequence identity of only 25%. Most importantly, upon structural overlap of the DraG/ADPr complex, the pose of ADPr fitted in the structure of ARH3 (PDB code 2FP0) with the distal ribose close to the two Mg2+ ions and without any steric conflict except for the slight re-orientation of Y149 during energy minimization (Fig. 1c, right panel). Thus we used the structural overlap to identify one potentially catalytically important water molecule and seven conserved amino acid residues with the following potential functions: holding water molecules in place (E41, D77), holding Mg2+ ion in place (D314, T317, E41), and binding of substrate (i.e., adenine and phosphate) (S148, H182, Y149) (Fig. 1c, right panel, and Supplementary Fig. 1d). To strengthen these findings, the binding of ARH3 to MARylated peptides was further studied by mutational analysis and by comparison with results of previously described ARH3

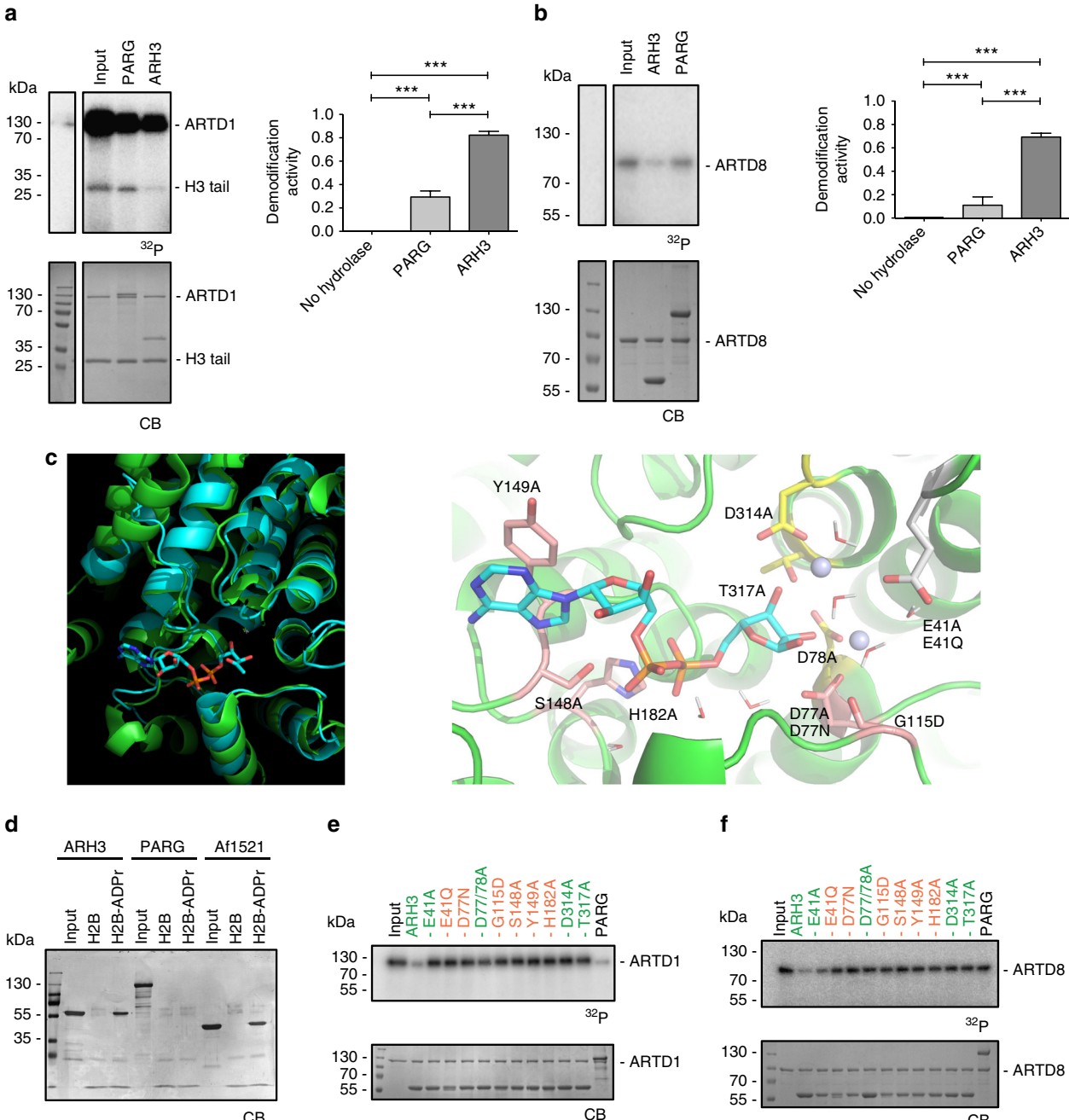

**Fig. 1** ARH3 has mono-ARH activity. **a** Left panel: Recombinant H3 histone tail was in vitro ADP-ribosylated using recombinant ARTD1 in the presence of $^{32}$P-labeled NAD$^+$. Equal fractions were left untreated (Input) or were treated with PARG or ARH3. Above: radioactivity exposure, below: Coomassie Blue-stained poly-acrylamide gel. Right panel: Quantification of **a** expressed as demodification activity (=reduction of the radioactive signal, normalized to amount of protein). Data represent means ± SEM for $n = 3$ independent experiments, ***$P < 0.0001$ as determined by ANOVA. **b** Left panel: Automodification of recombinant ARTD8 in the presence of $^{32}$P-labeled NAD$^+$ results in MAR-labeled ARTD8 (Input) that was subsequently treated with recombinant PARG or ARH3. Above: radioactivity exposure, below: Coomassie Blue-stained poly-acrylamide gel. Right panel: Quantification of **b** expressed as demodification activity (=reduction of the radioactive signal, normalized to amount of protein). Data represent means ± SEM for $n = 3$ independent experiments, ***$P < 0.0001$ as determined by ANOVA. **c** Left panel: Structural overlap of human ARH3 (green) and the DraG/ADPr complex (cyan). Right panel: Zoom in the active site of ARH3 with side chains of key residues which were mutated (labels). The binding mode of ADPr (carbon atoms in cyan) was obtained by energy minimization starting from the pose obtained by the structural overlap. **d** Coomassie Blue-stained membrane of pull-downs of GST-ARH3 (left) or His-PARG using the biotinylated peptides with (H2B-ADPr) and without (H2B) modification. Af1521 served as positive control. **e** ARTD1 automodified in the presence of $^{32}$P-labeled NAD$^+$ was subjected to demodification using WT and different ARH3 mutants. Red labels: mutants deficient in binding to H2B-ADPr, green labels: mutants retaining binding to H2B-ADPr. **f** ARTD8 automodified in the presence of $^{32}$P-labeled NAD$^+$ was subjected to demodification using WT and different ARH3 mutants. Red labels: mutants deficient in binding to H2B-ADPr, green labels: mutants retaining binding to H2B-ADPr

mutants[12,21]. To test the binding to a MARylated peptide, we established a pull-down assay using biotinylated synthetic ADP-ribosylated peptides with sequences derived from histone H2B tails (Supplementary Fig. 1e)[10,32]. As the synthesis of H2B modified at position 2 (Glu/E) could not be achieved, Glu was replaced by a Gln/Q[32]. This modification renders the peptide resistant to acyl migration and thus the linkage is not cleavable by ADPr hydrolases, which allows the analysis of binding abilities in the absence of hydrolysis[32]. Moreover, the binding assays were performed at 4 °C, a temperature at which ARH3 activity is greatly reduced (Supplementary Fig. 1f), strongly suggesting that ARH3's hydrolytic activity is absent during the binding assays. The unmodified and modified peptides were bound to streptavidin beads and incubated with either glutathione-S-transferase (GST)-tagged ARH3 or His-tagged PARG. After pull-down, Coomassie staining of the blotted membrane revealed a single 55 kDa band representing GST-ARH3 only when ADPr-modified peptides were used for the pull-down, similar to Af1521 known to be a strong binder of ADP-ribose[33]. PARG apparently neither bound the unmodified nor the modified peptide under the tested conditions (Fig. 1d). The data thus strongly suggest that ARH3, but not PARG, is able to stably bind to MARylated peptides. To substantiate that the above identified residues indeed play a role for either MAR-binding or hydrolase activity of ARH3, we generated different amino acid point mutants (e.g., E41A, E41Q, D77N, G115D, S148A, Y149A, H182A, D314A, and T317A) plus a double alanine replacement for D77 and D78. MAR-binding

assays using the above described pull-down approach with chemically modified H2B peptides revealed six residues to be crucial for binding of ARH3 to MARylated peptides (or overall ARH3 structure) (E41, D77, G115, S148, Y149, and H182) (Supplementary Fig. 2a). However, the E41A, the D77/78A as well as the D314A and T317A mutants retained their MAR-binding capacity. With respect to ARH3's enzymatic activity, demodification assays using ARTD1 automodified with radioactively labeled NAD$^+$ showed that all identified residues including the ones that seem to be dispensable for binding (i.e., see mutants E41A, D77/78A and D314A) are crucial for the PAR hydrolase activity of ARH3 (Fig. 1e), in line with previously published data[13]. Similar results were obtained when analyzing H3 tails modified by ARTD1 (Supplementary Fig. 2b). However, since the data with ARTD1-modified targets does not allow a conclusion about the MAR hydrolase activity per se but only about the combined PAR/MAR hydrolase activity of ARH3, the same ARH3 mutants were tested with automodified ARTD8, which acts as mono-ART and therefore solely catalyzes MARylation (Fig. 1f). These experiments revealed that most of the tested ARH3 mutants also lost their activity to demodify MARylated ARTD8. Only E41A retained weak activity, although not to the same extent compared to WT ARH3. In contrast, the E41Q mutant completely lost the mono-ADP-ribosyl-hydrolase activity, which is most likely due to its loss of ADP-ribose binding capacity.

In summary, while some of the tested ARH3 mutants were still able to bind the MARylated peptide, the D77, D78, G115, S148,

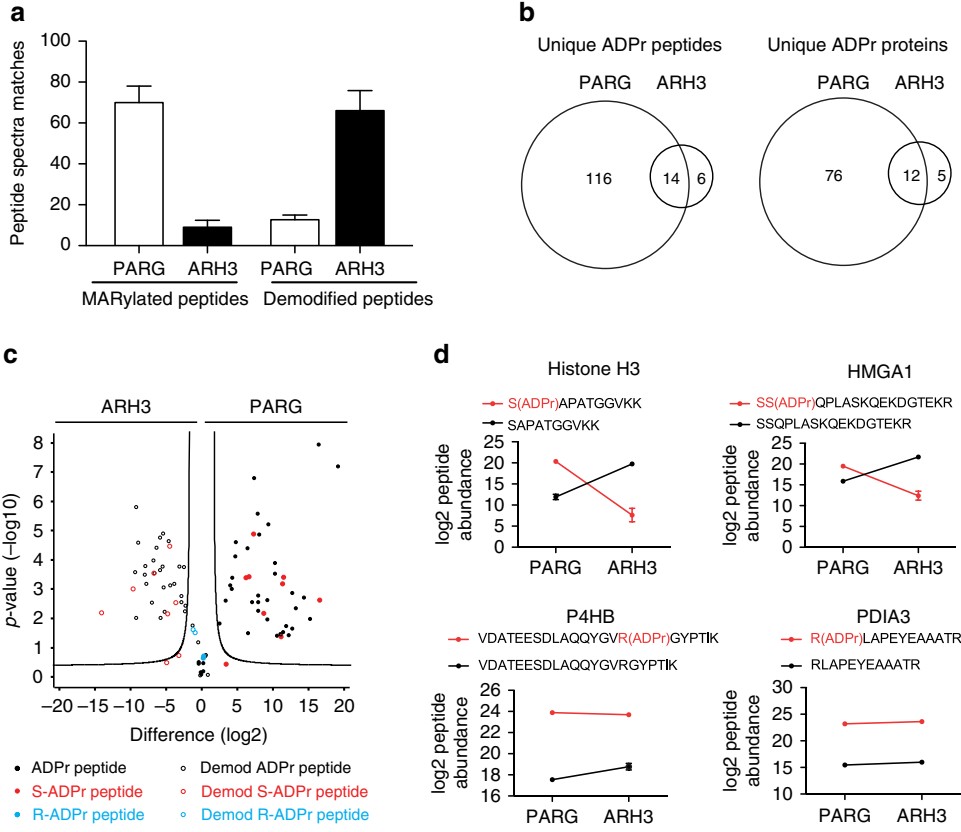

**Fig. 2** ARH3 mainly hydrolyzes ADP-ribosylated serines in vitro. **a** Number of ADP-ribosylated peptide spectra matches (PSMs) or demodified peptide spectra matches after PARG or ARH3 treatment. Data represent means ± SEM for $n = 3$ independent demodification experiments. **b** Venn diagrams of unique ADPr peptides and proteins. **c** Volcano plot of ARH3- and PARG-treated samples. "Unmodified peptides" are shown as open circles and "ADP-ribosylated peptides" as filled circles. ADP-ribosylation sites confirmed by EThcD spectra are annotated and color coded in red as S-ADPr and in blue as R-ADPr sites. ADP-ribosylated peptides with uncertain ADP-ribosylation site localization are shown in black. The black hyperbolic line represents a permutation-based false discovery rate (FDR) of 5% and a minimal fold change of 2. **d** Normalized abundance of individual Ser- and Arg-ADPr peptides after PARG or ARH3 treatment. Data represent means ± SEM for $n = 3$ independent demodification experiments

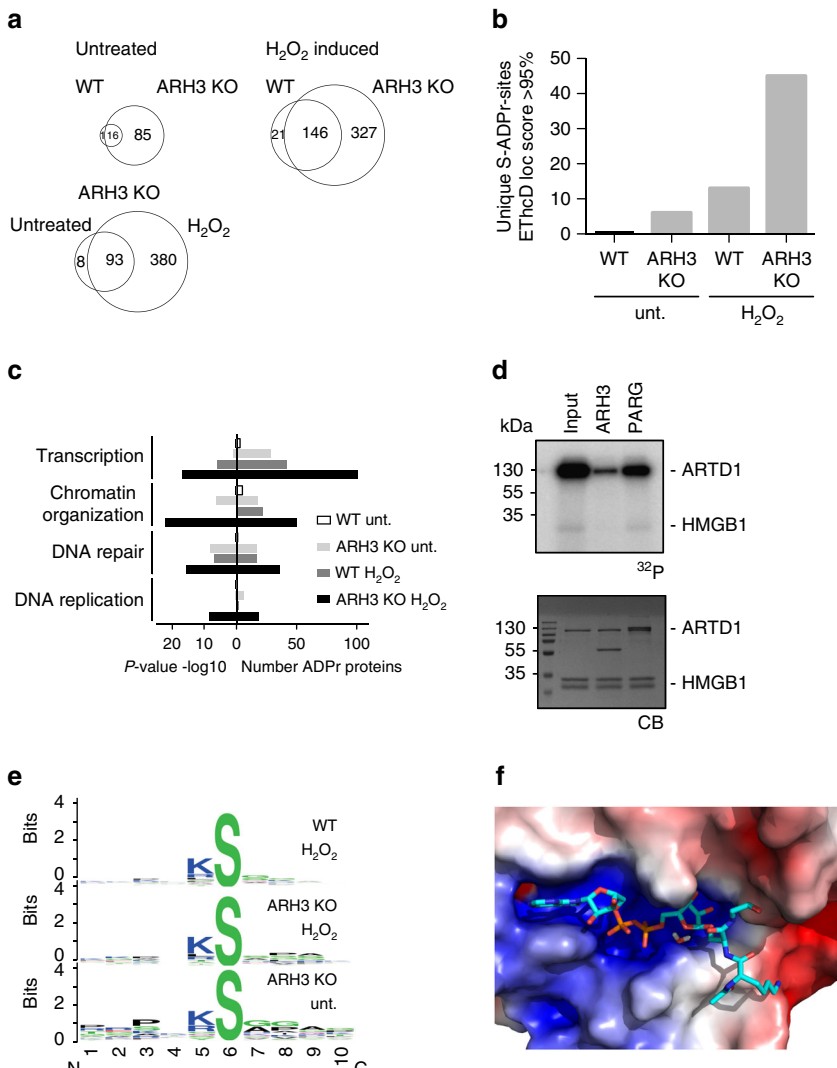

**Fig. 3** ARH3 regulates basal and hydrogen peroxide-induced serine ADP-ribosylation in vivo. **a** Venn diagrams of unique ADP-ribosylated peptides of wild type (WT) and ARH3 KO MEF cells under basal and $H_2O_2$-treated conditions. **b** Unique ADP-ribosylation sites detected by EThcD fragmentation in the different samples. **c** Gene ontology analysis of the identified ADPr-modified proteins using the PANTHER database. Shown on the left are the P-values and on the right the number of identified and annotated ADP-ribosylated proteins. **d** Validation of mono-ARH activity of ARH3 on the nuclear protein HMGB1. Recombinant HMGB1 was in vitro ADP-ribosylated using recombinant ARTD1 in the presence of $^{32}P$-labeled $NAD^+$. Equal fractions were left untreated (Input) or were treated with PARG or ARH3. Above: radioactivity exposure, below: Coomassie Blue-stained poly-acrylamide gel. **e** Motif searches for ADP-ribosylated peptides with a mascot site localization score >80% in MEF cells using Weblogo. **f** Energy minimized binding mode of an acetyl-KSG peptide with ADPr-Ser modification. The surface of ARH3 (including the binding-site magnesium ions) is colored according to electrostatic potential (on a scale of −5 to 5 kT/e). The positively charged amino group of the K side chain and the backbone amide groups point toward the region of the surface with negative potential

Y149, H182, D314, and T317 ARH3 mutants lost the ability to demodify both PARylated and MARylated target proteins, whereas the amino acid residue E41 proved to be slightly less important for the demodification of MARylated target proteins.

**ARH3 hydrolyzes serine ADP-ribosylation in vitro.** To address which ADPr acceptor sites can be demodified by ARH3, we applied a label-free quantification (LFQ) mass spectrometry (MS) approach, which was previously applied by our group[34]. We used $H_2O_2$-stressed HeLa cells as a model system to generate a broad array of ADP-ribosylated proteins[35]. To prevent ADP-ribosylation and de-ADP-ribosylation of proteins during lysis, tannic acid, which inhibits both PARG and ARH3 (Supplementary Fig. 2c) as well as PJ34 (a PARP inhibitor) were added

to the lysis buffer. Tryptic digest and subsequent Af1521 enrichment[35,36] resulted in a pool of PARylated and MARylated peptides, which was subjected to demodification by either recombinant ARH3 or PARG (Supplementary Fig. 2d). Subsequent shotgun liquid chromatography (LC)-MS/MS analysis of three biological replicates revealed that PARG treatment resulted in the identification of many spectra with MARylated peptides, while ARH3 treatment significantly reduced the quantity of spectra with MARylated peptides (Fig. 2a). Vice versa, the frequency of completely demodified peptides, i.e., the non-modified version of identified ADP-ribosylated peptides, increased upon ARH3 treatment (Fig. 2a). Most of the ADP-ribosylated peptides/proteins detected after PARG treatment were no longer observed upon ARH3 treatment (Fig. 2b), suggesting that indeed ARH3 is capable of fully removing PARylation and MARylation from

peptides. We performed LFQ of the ADP-ribosylated peptides and their non-modified counterparts based on the MS1 precursor peak area. This quantitative analysis confirmed that the MARylated peptides detected after PARG treatment were lost when the peptides were treated with ARH3 instead, which resulted in a gain of the corresponding unmodified peptides (Supplementary Fig. 2e). This data was acquired using HCD peptide fragmentation, which allows the identification of ADP-ribosylated peptides and the determination of the ADP-ribosylation sites, although we have recently reported that the accuracy of ADPr acceptor assignment is difficult when potential acceptor amino acids are located beside each other due to extensive fragmentation of the ADP-ribose moiety[10,35]. In contrast, ETD-based fragmentation and especially EThcD fragmentation is advantageous for this task and provides much higher accuracy and confidence for the localization of the ADPr acceptor site on the identified peptide. Therefore, to accurately annotate the modification sites for a part of the modified peptides, we combined the identified ADP-ribosylated peptides from this HCD-based MS measurement with the exact ADP-ribosylation site localization from a high-quality dataset that has recently been generated on the same biological setting using a combined HCD and EThcD approach[10] (Supplementary Data 1). To this end, we adopted the high-resolution site localization information by matching the here identified ADP-ribosylated peptides to the ones we previously reported in Bilan et al.[10]. To address which ADP-ribose acceptor site(s) are specifically demodified by ARH3, we compared the ARH3-treated samples vs the PARG treated samples and displayed the abundance of ADP-ribosylated peptides and their unmodified counterparts by a volcano plot (Fig. 2c). ARH3 demodified the majority of the Af1521 enriched PARylated/MARylated peptides, while only a small set of MARylated peptides could not be demodified by ARH3. Almost all monitored peptides with annotated serine-ADPr were significantly demodified by ARH3, while some of the non-demodified MARylated peptides contained arginine as ADPr acceptor site (Fig. 2c), providing strong evidence that ARH3 preferentially demodifies peptides with ADP-ribosylated serine residues. Since the mapped ADP-ribosylated peptides with ADPr acceptor sites that had been accurately assigned using the EThcD were still low in numbers, we also matched our data to high-scoring HCD fragmentation spectra and repeated the volcano plot analysis (Supplementary Fig. 2f). Although the site localization is more promiscuous with the HCD data, this analysis again strengthened our conclusion that ARH3 completely demodifies MAR/PARylated serine but not MARylated arginine.

One of the most prevalent protein groups found among these spectra to be demodified by ARH3 consisted of nuclear proteins including histones. To verify that ARH3 specifically demodifies serine ADPr acceptor sites in vivo, we carefully monitored, among the above described enriched peptide pools, two peptides with serine ADPr acceptor sites (i.e., H3 and HMGA1) that were recently published to be modified at these serine residues[37] and two peptides with known arginine-acceptor sites[35] (i.e., P4HB and PDIA3). Detailed analysis confirmed that the serine sites of H3 and HMGA1 were significantly demodified by ARH3, whereas the modified arginine ADPr acceptor sites of P4HB and PDIA3 remained equally abundant whether the peptides were treated with PARG or with ARH3 (Fig. 2d). Thus our data strongly indicates that ARH3 is a serine mono-ARH in vitro. It remains to be determined which residue(s) regulate the specificity of DraG and ARH1 (both arginine specific) and ARH3 (serine preference). Although ARH1 and ARH3 share amino acid sequence similarity and enzymatically crucial residues (D77 and D78), the modeling of the electrostatic

surface potentials shows substantial differences between the two enzymes (Supplementary Fig. 2g). The structural overlap of DraG with ARH3 revealed that G115 could be a possible candidate residue confering arginine specificity (Fig. 1c). However, mutating residue G115 of ARH3 to aspartate did not confer arginine-MAR-hydrolase activity when tested on CDTa-mediated ADP-ribosylated actin (Supplementary Fig. 2g).

**ARH3 reduces serine ADP-ribosylation in vivo.** With the above experiments, we showed that human ARH3 is a serine mono-ARH in vitro. Notably, human and mouse ARH3 share a high amino acid sequence identity. To address whether ARH3 contributes to the demodification of MARylated proteins in vivo, we thus mapped all ADP-ribosylated peptides (i.e., the ADP-ribosylomes) of WT and ARH3 KO MEFs from mice which both expressed PARG comparably (Supplementary Fig. 3a). Analyses of ADP-ribosylated peptides were performed according to a LC-MS/MS measurement tailored for the analysis of ADP-ribosylated peptides[10,35]. Importantly, samples were all treated with PARG before enrichment and during the MS measurement ADP-ribose fragment ions were used for the selection of modified peptides to produce high-quality HCD as well as EThcD spectra for the accurate localization of the ADP-ribose modification on the peptide sequence. Notably, the total quantity of ADP-ribosylated peptides derived from ARH3 KO MEFs tremendously increased compared to WT MEFs already under basal (i.e., untreated) conditions (Supplementary Fig. 3b). We found a roughly five-fold increase in the number of different ADP-ribosylated peptides, indicating that ARH3 strongly contributes to the ADP-ribosylation state of many proteins (Fig. 3a). Most of the identified proteins were modified at only one site, while few were modified at several sites (Supplementary Fig. 3c). ARH3 KO MEFs were earlier described to form PAR with different kinetics compared to their WT counterparts[15]. When analyzing the cellular ADP-ribosylome of ARH3 WT vs KO cells treated with 500 μM $H_2O_2$ for 60 min, we found an increased number of ADP-ribosylated peptides in both cell types as expected and described before for HeLa cells[34] but to a much larger extent in ARH3 KO cells (Fig. 3a and Supplementary Fig. 3b). Interestingly, the overlapping fraction of modified peptides comparing WT vs KO remained significant after $H_2O_2$ treatment, and many more ADP-ribosylation sites could be observed in the ARH3 KO cells (Fig. 3a). Furthermore, there was a substantial overlap of ADP-ribosylated peptides derived from $H_2O_2$-treated WT MEFs and untreated ARH3 KO cells (Supplementary Fig. 3d). Measurement of biological replicates confirmed the reproducibility of the biological finding and analysis pipeline (Supplementary Fig. 3e).

Utilizing the EThcD-generated ADP-ribosylated peptide spectra, we found that there is a significant and specific increase in the number of unique serine-ADP-ribosylated sites both in untreated and in $H_2O_2$-treated conditions comparing ARH3 KO to WT cells (Fig. 3b). Using EThcD spectra with a localization score of >95%, only peptides with modified serines could be identified. The inclusion of also HCD spectra, which are not ideal for ADPr site localization, led to the identification of more but also different sites (e.g., lysine residues) (Supplementary Fig. 3f). Gene ontology analysis revealed that most of the proteins whose ADP-ribosylation increased in ARH3 KO cells regulate DNA-associated processes in the chromatin context, although to a differential extent dependent on the treatment and the cell type analyzed (Fig. 3c). This is in agreement with the observation that a large part of ARH3 is associated with chromatin (Supplementary Fig. 3g). To confirm that the identified proteins are direct ARH3 targets, we biochemically validated the demodification of one candidate target, HMGB1. After in vitro ARTD1-dependent

ADP-ribosylation, ARH3, but not PARG, almost completely demodified HMGB1 (Fig. 3d).

Since it was reported that the amino acids surrounding an ADP-ribosylation site influence the binding of certain macro-domains[32], we analyzed the amino acid sequence of ARH3-targeted ADP-ribosylation sites. For this, we created sequence logos within a 13 amino acid window with the identified ADP-ribosylation site in the center. As previously observed by us and others[10,35], we could confirm a primitive KS and a lower abundant RS motif for a significant portion of serine-ADPr sites in untreated ARH3 KO cells and upon $H_2O_2$ treatment of ARH3 WT and KO cells (Fig. 3e), indicating that ARH3 is targeting the same ADP-ribosylation sites under basal and after $H_2O_2$ treatment. A list of all identified ADP-ribosylated proteins and peptides can be found in supplementary materials (Supplementary Data 2).

Finally, based on the observed sequencing motives (Fig. 3e) and modeling of ADPr in the putative ARH3-binding site, we added both a KSG and a RSG tripeptide to the ARH3-ADPr model, restraining each during energy minimization by a covalent bond between the serine side chain and the ADPr close to E41 (Fig. 3f and Supplementary Fig. 3h). Interestingly the side-chain termini of both lysine and arginine next to the modified serine residue were capable of adopting a conformation allowing association with a patch close to the postulated active site having a negative electrostatic potential, indicating that the neighboring amino acids of an ADPr acceptor sites might contribute to the binding and subsequent demodification by ARH3. Moreover, in this model, the modified serine side chain occupies a cavity lined by G115. Mutation of this residue to a larger residue such as aspartate indeed prevented binding (Supplementary Fig. 2a).

In summary, our results provide strong evidence that ARH3 is responsible for the demodification of MARylated as well as PARylated peptides modified at a serine residue in vivo with potentially important consequences on DNA-associated processes both in physiological and pathological conditions. This data puts ARH3 in line with PARG as the direct antagonist of nuclear ARTD-mediated ADP-ribosylation, with ARH3 having the potential to completely reverse serine protein ADP-ribosylation.

## Discussion

We report here that ARH3, in addition to its PAR-degrading activity, exhibits a serine mono-ARH activity. This is reflected by the fact that ARH3 demodifies MARylated ARTD8. Furthermore, we showed that ARH3 binds to MARylated peptides and we identified crucial amino acid residues responsible for binding and demodification of MARylated targets (Fig. 1). Using MS approaches, we showed that ARH3 demodifies serine acceptor sites (Fig. 2). Finally, we provided in vivo evidence for an important regulatory role of ARH3 in chromatin organization and gene expression both under physiological and pathological conditions (Fig. 3).

ARH3 was earlier described to bind free ADP-ribose with micromolar affinity and to efficiently demodify PAR but not mono-ADP-ribose-arginine, -cysteine, -diphthamide, or -aspar-agine bonds[12,13]. The observed demodification of H3 in Fig. 1a is most likely a combination of first demodification of PARylation and subsequently de-MARylation of H3 (since PARG reduces the modification to a lesser extent). This is in agreement with the observation that the mono-ART ARTD8 can be demodified by ARH3 but not PARG (Fig. 1b). The finding of increased frequency of fully demodified peptides using the in vitro approach with Af1521-enriched PARylated and MARylated peptides (Fig. 2) demonstrates, on the molecular level, that the chemical specificity of the mono-ARH activity of ARH3 is indeed that of a

glycohydrolase (and not, for instance, of a phosphodiesterase), since we found no left-over traces of the ADPr modification by MS. Moreover, our unbiased quantitative MS approach revealed that ARH3 is able to completely demodify ADP-ribosylated serine but not arginine residues (Fig. 2c, d). During the revision of the manuscript, Fontana et al. also reported on the mono-ARH specificity of ARH3 using an in vitro system of purified target proteins as well as WT and mutant ARTD1 constructs and showed that ARH3 reverses only serine mono-ADP-ribosylation[30]. It remains to be shown how the spe-cificities of DraG, ARH1, and ARH3 are controlled. Initial attempts failed to render ARH3 an arginine-specific ARH by mutating G115 to aspartate and rather prevented binding to the MARylated peptide, suggesting that other amino acids contribute to the specificity.

Interestingly, different viral macrodomains (e.g., Chikungunya virus, O'nyong'nyong virus, Sindbis virus, or Venezuelan Equine Encephalitis virus) have recently been reported to demodify ARTD8 in vitro[38], suggesting that potentially viral macrodomains might share the target specificity with ARH3. Whether ARTD8 is also modified in vitro at serine residues or at other acceptor sites (e.g., lyines or glutamic acid) and whether ARH3 is able to demodify other ADP-ribosylated residues at the proteome-wide level has to be analyzed further.

ARH3 did not completely demodify several MARylated target proteins (e.g., ARTD8 or HMGB1; Fig. 1b, remaining modifica-tion of 30%), indicating that either the tested in vitro conditions were sub-optimal, that the amino acid sequence might contribute to the ability of the modified serine to be demodified, or that additional amino acids (e.g., arginines) are MARylated in vitro, which cannot be demodified by ARH3. Comparably, PARG and ARH3 were both unable to reduce the automodification of ARTD1 to undetectable levels (Fig. 1a), raising the question as to what renders ARTD1 resistant to full demodification. This find-ing is, however, in agreement with our earlier observation that ARTD1 is found to be modified already under basal conditions in different cell types[34]. The inability to demodify ARTD1 could be due to steric hindrance since the detected acceptor sites span a rather short domain of ARTD1 and might inhibit full demodifi-cation or the acceptor-site linkage might not be cleaved (e.g., lysine modification) due to promiscuous enzymatic activity in vitro.

The molecular architecture of ARH3 constitutes the archetype of an all-α-helical protein fold and provides insights into the reversibility of protein ADP-ribosylation[13]. Two $Mg^{2+}$ flanked by highly conserved amino acid residues pinpoint the active-site crevice. The structural overlay of DraG and ARH3 provided insights into a conserved catalytic cleft containing two potentially catalytically important water molecules. The known D77/78A mutants have been described before to be important for the binding of the water molecule. Interestingly, neither the water nor the $Mg^{2+}$ ion seem to be important for the binding of the ADP-ribose to ARH3. Notably, the mutation analysis also provided evidence that the PAR- and MAR-hydrolase activity are depen-dent on a large set of common residues. Interestingly, only the mutation of E41 to Q but not A affected the demodification of MARylated ARTD8. Based on preliminary modeling of a lysine- and an arginine-containing ADP-ribosylated serine tripeptide, it is suggested that E41 interacts not only with the magnesium cations (via binding site water molecules) but possibly in addition also with the tripeptide backbone. Mutation of E41 to alanine is anticipated to disturb the water structure surrounding the mag-nesium cations, with resulting loss of catalytic activity. By con-trast, mutation of E41 to glutamine may be hypothesized to interfere with the binding of the ADP-ribosylated serine back-bone and its flanking amino acid residues.

ARH3 was localized to mitochondria but is also present in the nucleus and the cytosol[12,39]. Our own data provide evidence that a substantial amount of ARH3 is associated with chromatin (Supplementary Fig. 3g). Interestingly, most of the identified potential ARH3 targets under basal conditions were nuclear protein previously assigned to DNA-associated processes (Fig. 3c), indicating that ARH3 is predominantly functionally relevant in the nucleus under the tested conditions. The detection of ADP-ribosylated peptides in ARH3 KO MEFs already under basal conditions strongly suggest that ARH3 contributes to the demodification of proteins under basal conditions, although we can not exclude that other enzymes might contribute to the observed increase of the ADP-ribosylome as well (Fig. 3a). It is currently not known which cellular ARTD is responsible for ADP-ribosylation under basal conditions. Since many of the identified nuclear proteins were modified at serines upon treatment with $H_2O_2$, and $H_2O_2$ is known to mainly induce ARTD1[40], it is tempting to speculate that the proteins identified in ARH3 KO cells under unstressed conditions are modified by ARTD1 as well. In the presence of ARH3, these proteins might be immediately demodified and thus not detectable. Whether the same known molecular mechanisms (e.g., spontaneous DNA lesions, DNA replication stress) induce ARTD1/2 activation under unstressed conditions needs to be further clarified. Moreover, due to the lack of MS-based methods that discriminate between MARylated and PARylated proteins, we can currently not exclude that the identified ADP-ribosylated peptides in ARH3 KO cells are not only MARylated but, to a certain extent, also PARylated. However, given that we are not able to detect any signal by immunofluorescence using an antibody against PAR strongly indicates that the majority of these peptides are indeed likely MARylated (Supplementary Fig. 4a). This conclusion is further supported by the observation that the knockdown of PARG leads to PAR formation under basal conditions that can be observed by immunofluorescence using an anti-PAR antibody[41,42]. The in vivo experiment also revealed that at least two types of acceptor sites (i.e., serine and arginine) are mainly modified in vivo after $H_2O_2$ treatment and that, consistent with the in vitro experiments (Fig. 2a, b), ARH3 was demodifying the ADP-ribosylated serine but not the modified arginine residues. Interestingly, proteins ADPr-modified at arginines were described to localize to the ER[34], while the proteins modified at serines rather localize to the nucleus. It is tempting to speculate that different ARTs are responsible for the identified ADPr acceptor sites.

Besides PARG, ARH3 also participates in nuclear and cytoplasmic PAR degradation under oxidative stress conditions, thus providing a reason why ARH3 KO MEFs were more susceptible to $H_2O_2$-induced cytotoxicity compared to WT MEFs[15]. In view of our findings, ARH3 might, under oxidative stress conditions, not only protect cells by degrading PAR but also by demodifying MARylated proteins. It remains to be investigated under which cellular conditions ARH3 would preferably demodify MARylated vs PARylated target proteins. This certainly also depends on the binding affinity of ARH3 to PARylated and MARylated targets. Along this line, the identified lysine or arginine residue located next to the modified serine residue (Fig. 3e, f) might significantly contribute to the binding of modified peptides to ARH3 and thus favor this type of activity.

The functional relevance of the newly identified protein ADP-ribosylation sites and the potential crosstalk with other modifications (e.g., serine phosphorylation) need to be further investigated. Considering the structural and potentially functional differences between PARG and ARH3, ARH3 might be an interesting therapeutical target. Specific inhibitors of ARH3 activity might help resolving its biological function in cells both under physiological and pathophysiological conditions.

In summary, we show that the PAR hydrolase ARH3 also has serine mono-ARH activity and that ARH3 activity very likely contributes to chromatin organization and DNA-associated processes, both in physiological and pathological conditions.

## Methods

**Cloning and protein purification.** Human ARTD1 was cloned, expressed, and purified as previously described[43]. GST-histone tail fusion proteins (GST-H3 and GST-H2B tail) were cloned and purified as previously described[44]. WT human ARH1 and ARH3 were cloned using primers (Microsynth) to amplify the sequence from a purchased cDNA clone (BioCat) by PCR and cloned into pGEX6P-3 using restriction enzymes BamHI and XhoI (NEB). ARH3 mutants were cloned by site-directed mutagenesis with fragment and overlap PCRs. The catalytic domain of mouse ARTD8 (residues 1216–1817) was cloned into pGEX6P-3 using restriction enzymes BamHI and SalI (NEB). Sequencing of plasmids was performed at Microsynth.

Plasmids were transformed into BL21 *Escherichia coli*, and protein expression was induced by adding 1 mM IPTG at $OD_{600}$ 0.4–0.6 for 3 h at 30 °C. Batch purification of GST-tagged proteins was performed using glutathione sepharose 4B beads (GE Healthcare) according to the manufacturer's manual. Expression and purification of all recombinant proteins was analyzed by sodium dodecyl sulfate-polyacrylamide gel electrophoresis (SDS–PAGE) followed by Coomassie staining.

**In vitro ADP-ribosylation assay with recombinant proteins.** To obtain ADP-ribosylated ARTD1, H3 and H2B tails, and HMGB1, recombinant proteins were incubated in reaction buffer RB (50 mM Tris-HCl pH 7.4, 4 mM $MgCl_2$ and 250 μM dithiothreitol (DTT)) at a ratio 1:3 (ARTD1 to histone tail) with 100 nM [$^{32}$P] NAD$^+$ (Perkin Elmer) and 200 nM of double-stranded annealed 40 bp long oligomer (5′-TGCGACAACGATGAGATTGCCACTACTTGAACCAGTGCGG-3′) for 15 min at 37 °C. ADP-ribosylation was stopped by adding 10 μM PJ34.

Automodification of ARTD8 was carried out in RB with 150 nM [$^{32}$P]NAD$^+$ and 10 μM cold NAD$^+$ for 1 h at 37 °C. The reaction was stopped by filtering through an Illustra MicroSpin G-50 column (GE Healthcare) according to the manufacturer's manual.

ADP-ribosylation of actin was performed as described earlier[45]. Briefly, 2 μg β/γ actin (Cytoskeleton Inc.) was incubated with 50 ng CDTa in the presence of 100 nM [$^{32}$P]NAD$^+$, 150 μM cold NAD$^+$ and reaction buffer (5 mM HEPES, pH 7.5, 0.1 mM $CaCl_2$, 0.5 mM NaAc, 0.1 mM ATP) at 37 °C. The reaction was stopped by filtering through a G50 column.

**In vitro de-ADP-ribosylation assay.** Demodification reactions were performed in RB. Unless otherwise stated, 10 pmol automodified ARTD1 and 30 pmol trans-modified histone tail were incubated with 10 pmol PARG or ARH3 for 15 min at 37 °C. For demodification of ARTD8, 30 pmol automodified recombinant protein were incubated with 30 pmol PARG or ARH3 for 1 h at 37 °C. Chemical demodification of ADP-ribosylated actin was performed by addition of 0.5 M hydroxylamine and incubation for 15 min at 37 °C or overnight incubation at 37 °C with the respective hydrolases. Reactions were stopped by adding SDS-loading buffer, with subsequent boiling at 95 °C for 5 min. Samples were run on an SDS-PAGE gel, stained with Coomassie Blue, photographed, destained, and exposed on phosphorscreens overnight or up to 1 day. Images were taken with a Typhoon FLA 9400 phosphorimager (GE Healthcare). Signal intensities were analyzed and quantified using ImageJ. For quantification of the MAR hydrolase activity, the signal intensity (radioactivity and Coomassie) for every stained band was determined using ImageJ. The radioactivity signal $R$ normalized to the respective Coomassie signal $C$ results in the specific modification signal $S$.

$$S = \frac{R}{C}$$

The specific activity $A$ of the demodifying enzyme is given by

$$A = 1 - \frac{S_X}{S_i}$$

where $S_X$ is the specific modification signal of a given condition (enzyme X) and $S_i$ is the specific modification signal of the input sample.

**Pulldown using biotinylated ADP-ribosylated H2B peptide.** To test the binding of ARH3 and PARG, biotinylated ADP-ribosylated or non-modified H2B peptides were used. Residue 2 of the peptide (originally an E), was changed to Q in order to chemically attach ADPr, since solid-phase synthesis of an ADP-ribosylated N-terminal tetrapeptide of H2B could not be achieved owing to side reactions initiated by migration of the 1-O-glutamyl moiety when an E was present. In contrast, Q is resistant to acyl migration and thus the so built amide linkage is not cleavable by ADPr hydrolases[32]. Peptides were bound to streptavidin sepharose high-performance beads (GE Healthcare). For each pull-down, 5 μl of beads were primed by washing three times in binding buffer (50 mM NaCl, 50 mM Tris-HCl pH 8, 0.05% NP-40) and incubated overnight at 4 °C in 1 ml binding buffer with 5 μg of the modified or unmodified peptide. Beads were washed with 1 ml incubation

buffer (0.1% NP-40, 1× protease inhibitor cocktail (Roche), 50 mM Tris-HCl pH 8, 1 mM DTT, 50 mM NaCl). Thirty pmol of recombinant protein were incubated with the beads and 1 ml incubation buffer for 3 h at 4 °C. Subsequently, the beads were washed three times with incubation buffer and once with 1 ml cold phosphate-buffered saline (PBS) before addition of SDS-loading buffer, sample boiling, and western blotting.

**Cells.** MEFs from $Arh3^{+/+}$ (WT) and $Arh3^{-/-}$ (KO) mice were cultured as previously described[15]. HeLa, HEK, and U2OS cells were originally purchased from ATCC and cultured in Dulbecco's modified Eagle's medium (DMEM) supplemented with 5% penicillin/streptomycin and 10% fetal calf serum.

**Enrichment of ADP-ribosylated peptides from cell lysate.** For the PARG/ARH3 peptide demodification assay, HeLa cells were treated with 1 mM $H_2O_2$ in PBS containing 1 mM $MgCl_2$ and further processed as described previously[35] with the following alterations: Tannic acid (75 μM) was added to the lysis buffer and PARG treatment was omitted after tryptic digestion. For every condition, the macrodomain enrichment was performed in triplicates on 7 mg peptides. The eluted MARylated and PARylated peptides were reconstituted in 50 mM Tris–HCl, pH 8, 10 mM $MgCl_2$, 250 mM DTT, and 50 mM NaCl. Forty-five pmol ARH3 or PARG were added to the peptide mixture and the demodification reactions were performed at 37 °C for 1 h. Subsequently, the mixture was filtered through a Microcon-30 cut off centrifugal filter (Merck Millipore), acidified with TFA and desalted on reverse phase C18 StageTips.

The MEF WT and ARH3 knockout cells were either left in DMEM for 1 h or treated with 500 μM $H_2O_2$ for 1 h in DMEM. After one wash with PBS, cells were scraped and lysed by adding 6 M guanidine-hydrochloride (Gnd-HCl), 5 mM tris (2-carboxyethyl)phosphine, 10 mM 2-chloroacetamide, and 100 mM Tris pH 8, 95 °C[46,47]. The samples were diluted with 25 mM Tris, pH 8, and digested with trypsin (Promega). The peptide mixture was treated with PARG to obtain only MARylated peptides, and the peptides were enriched using a macrodomain affinity pull-down as described previously[35].

**Liquid chromatography and MS analysis.** MS analysis for the recombinant PARG/ARH3 peptide demodification assay was performed on an Orbitrap Q Exactive mass spectrometer (Thermo Fisher Scientific) coupled to a nano EasyLC 1000 (Thermo Fisher Scientific). The peptides were loaded onto a self-made column (75 μm × 150 mm), which was packed with reverse-phase C18 material (ReproSil-Pur 120 C18-AQ, 1.9 μm, Dr. Maisch GmbH). Solvent compositions in channels A and B were 0.1% formic acid in $H_2O$ and 0.1% formic acid in acetonitrile, respectively. The peptides were separated at a flow rate of 300 nL/min by a 130 min elution gradient protocol from 2 to 25% B in 100 min, from 25 to 35% B in 10 min, from 35 to 95% in 10 min, and 95% B for 10 min.

The mass spectrometer was set to acquire full-scan MS spectra (300–1700 $m/z$) at a resolution of 70,000 after accumulation to an automated gain control target value of $3 \times 10^6$. Charge state screening was enabled, and unassigned charge states and single charged precursors were excluded. Ions were isolated using a quadrupole mass filter with a 2 $m/z$ isolation window. A maximum injection time of 250 ms was set. HCD fragmentation was performed at a normalized collision energy of 25%. Selected ions were dynamically excluded for 30 s.

The identification of ADP-ribosylated peptides from MEF cells was performed on an Orbitrap Fusion Tribrid mass spectrometer (Thermo Fisher Scientific), coupled to a nano EasyLC 1000 liquid chromatograph (Thermo Fisher Scientific). We applied an ADP-ribose product-dependent method called HCD-PP-EThcD as described in ref. [10]. Briefly, the method includes high-energy data-dependent HCD, followed by high-quality HCD and MS when/MS when two or more ADP-ribose fragment peaks (136.0623, 250.0940, 348.07091, and 428.0372) were observed in the HCD scan. A detailed description of the MS parameters can be found in ref. [10]. Solvent compositions in channels A and B were 0.1% formic acid in $H_2O$ and 0.1% formic acid in acetonitrile, respectively.

Peptides were loaded onto an Acclaim PepMap 100 (Thermo Scientific) trap column, 75 μm × 2 cm, packed with C18 material, 3 μm, 100 Å, and separated on an analytical EASY-Spray column (Thermo Scientific, 75 μm × 500 mm) packed with reverse-phase C18 material (PepMap RSLC, 2 μm, 100 Å). Peptides were eluted over 110 min at a flow rate of 300 nL/min. An elution gradient protocol from 2 to 25% B, followed by two steps, 35% B for 5 min, and 95% B for 5 min was used.

**MS data analysis.** MS and MS/MS spectra were converted to Mascot generic format (MGF) by use of Proteome Discoverer, v2.1 (Thermo Fisher Scientific, Bremen, Germany). When multiple fragmentation techniques (HCD and EThcD) were utilized, separate MGF files were created from the raw file for each type of fragmentation. MGF files were further processed as described[10]. The MGFs resulting from measurement following the peptide demodification assay were searched against the UniProtKB human database (taxonomy 9606, version 20140422), which included 35,787 Swiss-Prot entries; 37,802 TrEMBL entries; 73,589 decoy hits; and 260 common contaminants. The MGFs resulting from the MEF cell measurements were searched against the UniProtKB mouse database (taxonomy 10090, version 20160902), which included 24,905 Swiss-Prot entries; 34,616 TrEMBL entries; 59,783 decoy hits; and 262 common contaminants.

Mascot 2.5.1.3 (Matrix Science) was used for peptide sequence identification with previously described search settings[48] and some modification for the EThcD searches[10]. Enzyme specificity was set to trypsin, allowing up to four missed cleavages. The ADP-ribose variable modification was set to a mass shift of 541.0611, with scoring of the neutral losses equal to 347.0631 and 249.0862. The marker ions at $m/z$ 428.0372, 348.0709, 250.0940, and 136.0623 were ignored for scoring. S, R, K, D, and E residues were set as variable ADP-ribose acceptor sites. Carbamidomethylation was set as a fixed modification on C and oxidation as a variable modification on M. Peptides are considered correctly identified when a Mascot score >20 and an expectation value <0.05 are obtained. For the ADP-ribosylation site analyses, peptides identified with EThcD fragmentation, having a mascot localization score >95% were used if not stated otherwise.

To perform a LFQ based on the MS1 precursor peak area of the identified peptides in the peptide demodification assay, Progenesis QI software (v. 3.0.6039.34628, Nonlinear Dynamics, Purham, NC) was applied. Raw data were imported into Progenesis and aligned based on the MS1 peak retention time. All samples were normalized based on the total signal intensity to account for sample loading variations. The obtained results were exported as MGF and searched with Mascot as indicated above. The Mascot search results were imported into the Scaffold software (v.4.7.2) and filtered for protein and peptide false discovery rate (FDR) values of 0.01. When multiple precursors were observed for the same peptide, the values were summed up to obtain the total level of the peptide. In order to ensure the correct assignment of the ADP-ribose localization on the peptide, we compared the here identified ADP-ribosylated peptides to our high-quality EThcD dataset published in ref. [10]. The so-called demodified peptides were obtained by matching the ADP-ribosylated peptides identified in this experiment to their identified non-modified counterparts.

For the ADP-ribosylation site motif analysis, we used Weblogo[49], including ADP-ribosylated peptides identified with a mascot localization score >80% and a sequence window of 6 amino acids around the modified site. Volcano plot analysis of the quantified ADP-ribosylated peptides and their non-modified counterparts was performed using two-sample testing in Perseus, with a permutation-based FDR of 5% and minimal fold change of $2^{50}$. Gene ontology analysis was performed using the PANTHER data base[51].

**Gene expression.** MEF cells were washed with PBS before performing RNA extraction with the NucleoSpin RNA II Kit (Macherey-Nagel). RNA was quantified with a NanoDrop (Thermo Fisher Scientific) and reverse-transcribed according to the supplier's protocol (High Capacity cDNA Reverse Transcription Kit, Applied Biosystems). Quantitative real-time polymerase chain reactions were performed with KAPA SYBR fast (Kapa Biosystems) and a Rotor-Gene Q 2plex HRM System (Qiagen).

**Chromatin extraction.** Cells were washed with 1 ml ice-cold PBS, collected, centrifuged at 1000×g for 5 min at 4 °C and washed twice more with PBS. The pellet was resuspended in three volumes chromatin extraction buffer (200 mM NaCl, 10 mM HEPES pH 8, 3 mM $MgCl_2$, 0.5 % Triton X-100, 1x protein inhibitor cocktail (Roche)) and incubated rolling for 30 min at room temperature. Centrifugation at 10,000×g for 10 min at 4 °C gave rise to the soluble fraction (supernatant). The pellet was resuspended with an equal amount of chromatin extraction buffer. The soluble fraction was sonicated once for 30 s, the chromatin fraction three times for 30 s. Protein concentration was measured using a Bradford assay (Biorad), and samples were subjected to SDS-PAGE and western blotting.

**Western blotting.** For western blotting analysis, proteins were separated by SDS-PAGE, and bands were visualized using the Odyssey infrared imaging system (LI-COR). Antibodies used for western blotting were anti-tetra-HIS (1:1000, Qiagen, #34670), anti-GST-Z5 (1:1000, Santa Cruz, #sc-459), anti-Tubulin (1:10,000, Sigma, #T6199), anti-H3 (1:5000 Abcam, #ab1791), anti-ARH3 (1:1000, custom made, Genosphere Biotech), IRDye 800CW goat anti-rabbit IgG (1:15,000, LI-COR, P/N 925-32211), and IRDye 680RD Goat anti-Mouse IgG (1:15,000, LI-COR, P/N 925-68070). Molecular weights are indicated by the PageRuler Plus Prestained Protein Ladder (Thermo Scientific). Uncropped western blottings are displayed in Supplementary Fig. 4.

**Modeling.** Sequence alignments of human ARH3 and *R. rubrum* DraG (Uniprot identifiers Q9NX46 and P14300, respectively) were performed using the Clustal Omega program on the Uniprot webserver[52,53]. The WITNOTP program (molecular modeling software developed by A. Widmer at Novartis AG, Basel) was used to protonate all protein and ligand structures and water molecules. In particular, the side chain of His182 of ARH3 was positively charged for all minimizations as it is located next ot the phosphate groups of ADPr-Ser. WITNOTP further served to align protein structures based on their Cα atoms (using the option STRUCTAL-3), to modify selected atom coordinates and to modify the ADP ribose ligand by addition of heavy atoms. All minimizations were performed with CHARMM[54] using the CHARMM36 forcefield for the protein atoms, water molecules, and magnesium atoms and the CHARMM general forcefield for each of the ligands[55]. Electrostatic potential surfaces, following application of the pdb2pqr software[56,57], were calculated with the Adaptive Poisson-Boltzmann Solver software package[58].

Figures were created with Pymol (The PyMOL Molecular Graphics System, Version 1.8 Schrödinger, LLC).

**Immunofluorescence**. MEF cells were grown on coverslips and treated with $H_2O_2$ in DMEM for the indicated times, and immunofluorescence using a homemade 10H antibody was performed as described[59] with the following deviations. The antibody was diluted 1:250, and DNA was stained in a separate step using Hoechst 33258 (Sigma) before mounting.

**Data availability**. The mass spectrometric proteomics data have been deposited to the ProteomeXchange Consortium via the PRIDE[60] partner repository with the dataset identifier PXD008083 and 10.6019/PXD008083. Additional data are available from the authors upon request.

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

## Acknowledgements

We thank Monika Fey (University of Zurich) for the expression and purification of recombinant human ARTD1 as well as PARG and Tobias Suter (University of Zurich) for helpful discussion and providing editorial assistance. CDTa was kindly provided by Klaus Aktories (University of Freiburg). We thank Paolo Nanni and Peter Gehrig (FGCZ, University of Zurich) for technical support for the MS measurements. J.M. and J.K. were supported by the Intramural Research Program, NIH, NHLBI. Research in the group of A.C. is funded by the Swiss National Science Foundation (SNF 315230_149897). ADP-ribosylation research in the laboratory of M.O.H. is funded by the Canton of Zurich and the Swiss National Science Foundation Grants (SNF 310030_157019) as well as the Forschungskredit from the University of Zurich (to M.L.).

## Author contributions

J.A. and M.O.H. planned the experiments. J.A. performed the experiments and analyzed the results. M.L. and K.N. performed mass spectrometric analysis and E.F. and A.C. the structural studies. R.F. performed H2B-peptide pulldowns. J.K. and J.M. provided wild type and ARH3 KO MEFs. H.A.V.K. and D.V.F. synthesized the modified H2B peptides. M.O.H. supervised the experiments and provided overall guidance; M.O.H., J.A., and M.L. wrote the manuscript.

## Additional information

**Competing interests:** The authors declare no competing financial interests.

