## [Peer Review File · Nature Communications]

PEER REVIEW FILE

Reviewers' comments:

Reviewer #1 (Remarks to the Author):

Although our understanding of the writers and erasers for protein Poly-ADP-ribosylation (PARylation) has greatly advanced in the last several years, the enzymes that regulate the synthesis and degradation of Mono-ADP-ribosylation (MARylation) are still poorly characterized. In particular, the erases that remove Ser MARylation remain elusive. In this study, the authors provided evidence suggesting that ARH3 could be a relevant enzyme that catalyzes the catabolism of Ser MARylation. Specifically, they showed that in vitro, ARH3 binds to and demodifies MARylated peptides and proteins. Using structural modeling, they identified residues within the catalytic center that are critical for its binding and hydrolyzing activities. Finally, they used a label-free quantitative proteomic approach, and characterized how the in vivo ADP-ribosylome differed between WT and ARH-KO MEFs, under both basal and oxidative DNA damage conditions. These findings fill in a gap in our knowledge with potentially important implications for the regulation of basal and stress-induced ADP-ribosylome.

Specific comments:

- (1) Page 2, what are the relative strengths in PAR-demodifying activity between ARH3 and PARG? The authors need to generate ARH3 and PARG single knockdown, as well as double-knock down cells and measure basal levels of and stress-induced PARylation.
- (2) Page 11, the authors need to provide evidence supporting that trans-modified H3 is a MARylated protein. Even though ARTD8 is known to be a mono-ART, the authors need to show what is the MARylated residue (i.e. D/E, K, or S)? Then they need to use mass spectrometry to follow the deMARylation of these residues, and demonstrate that ARH3 is a Ser-specific deMARylating enzyme.
- (3) Page 11, DraG needs to be consistent within the manuscript (all upper-case).
- (4) Page 11, the sequence of the H2B peptide is wrong. The second amino acid (with ADP-ribosylation) should be Glu instead of Gln. Furthermore, the authors later claim that ARH3 specifically demodifies Ser-ADP-ribosylation. Why did they test ARH3 against a peptide that is ADP-ribosylated on Glu (or Gln)?
- (5) Page 12, ARH3 and PARG need to be tested in one single experiment in order to assess their

relative affinity for H2B peptides. The authors also need to elaborate on how they decouple the binding activity from the hydrolyzing activity? Once ARH3 binds to an ADP-ribosylated peptide, it is expected that ARH3 will rapidly hydrolyze its ADP-ribosylation moiety. This will complicate the interpretation of the resulting data.

(6) Page 13, line 391-394, the discussion here is confusing. They need to be re-worded.

(7) Page 13, line 410-415. the way Figure 2C is illustrated is very confusing. How did the author extract the data for PARG vs ARH3 treated samples, and how the data were cross-referenced? What is each point compared to? what are the S0 values? Also, the authors need to list the distribution of MARylation on each amino acid.

(8) Page 15, the authors claim that ARH3 demodifies Ser-ADPr residues in a KS motif. Why did they then use a GSK peptide, instead of GKS peptide when modeling the binding between this ADP-ribosylated peptide and ARH3 active site. Can the authors mutate the K in a substrate and show that this abolishes its binding to ARH3?

Reviewer #2 (Remarks to the Author):

I have carefully read the manuscript titled "Proteomic analyses identify ARH3 as a serine mono-ADP-ribosylhydrolase". Although the area of research is significant and the results interesting, I do not feel the results are of wide enough impact, nor the methods innovative enough to justify publication in Nature Communications. I would suggest a more archival journal for this work.

The main methods of the paper are described as mutagenesis, modeling (spelled modelling in abstract and modeling in methods, suggesting different authors wrote separate sections) and a mass spec technique to show in vivo function, both basal and oxidative stress initiated. Numerous mutants were identified that had an impact on activity - a pedestrian result. Not sure what modeling was used for. The methods described modeling as protein sequence alignment, protonation and MM minimization (CHARMM). This is not innovative and it is not even modeling. And the mass spec technique was reported in ref 42, pointing out the lack of innovation here as well.

Additionally, the manuscript needs attention to abbreviations, grammar and multiple typos. A careful proofing before resubmission is advised.

Response to Reviewer's Comments on Manuscript NCOMMS-17-13941

We thank the reviewers for evaluating our manuscript and for the constructive criticism. The reviewers' suggestions helped to improve the quality and structure of the manuscript.

Below, we have outlined the specific changes made to the manuscript. Furthermore, we provide a detailed point-by-point response to each of the reviewer's comments. Changes in the manuscript have been **highlighted in yellow**. Minor changes (like typos, grammatical errors and slight changes in wording) were not highlighted.

Summary of changes made to the figures and tables:

Main Figures	Suppl. Figures
Fig. 1a	Fig. S1a
Fig. 1b	Fig. S1b
Fig. 1c	Fig. S1c
Fig. 1d, revised	Fig. S1d
Fig. 1e	Fig. S1e
Fig. 1f	Fig. S1f, revised
	Fig. S2a
Fig. 2a	Fig. S2b
Fig. 2b	Fig. S2c
Fig. 2c	Fig. S2d
Fig. 2d	Fig. S2e
	Fig. S2f
	Fig. S2g, revised
Fig. 3a, revised	Fig. S2h
Fig. 3b	
Fig. 3c	
Fig. 3d	Fig. S3a
Fig. 3e	Fig. S3b
Fig. 3f, revised	Fig. S3c
	Fig. S3d, new
	Fig. S3e
	Fig. S3f
	Fig. S3g
	Fig. S3h, new

Point-by-point response to the questions raised by the reviewer:

Reviewer 1

Although our understanding of the writers and erasers for protein Poly-ADP-ribosylation (PARylation) has greatly advanced in the last several years, the enzymes that regulate the synthesis and degradation of Mono-ADP-ribosylation (MARylation) are still poorly characterized. In particular, the erasers that remove Ser MARylation remain elusive. In this study, the authors provided evidence suggesting that ARH3 could be a relevant enzyme that catalyzes the catabolism of Ser MARylation. Specifically, they showed that in vitro, ARH3 binds to and demodifies MARylated

peptides and proteins. Using structural modeling, they identified residues within the catalytic center that are critical for its binding and hydrolyzing activities. Finally, they used a label-free quantitative proteomic approach, and characterized how the in vivo ADP-ribosylome differed between WT and ARH-KO MEFs, under both basal and oxidative DNA damage conditions. These findings fill in a gap in our knowledge with potentially important implications for the regulation of basal and stress-induced ADP-ribosylome.

Response: We thank the reviewer for the overall positive evaluation of our work.

1. Page 2, what are the relative strengths in PAR-demodifying activity between ARH3 and PARG? The authors need to generate ARH3 and PARG single knockdown, as well as double-knock down cells and measure basal levels of and stress-induced PARylation.

Response: Various labs have previously studied the PAR-degrading function of PARG and ARH3. We reorganized the existing text and included a new paragraph in the introduction (page 3) where we reference relevant studies, with focus on H₂O₂-induced stress conditions. Furthermore, in Supplementary Figures 1b and 1c, we compare PARG and ARH3 side-by-side. The results show that after 7-10 min the demodification reaction is completed (Suppl. Figure 1c) and that PARG and ARH3 have comparable specific molar activities (Suppl. Figure 1b).

Moreover, since this manuscript focuses on the identification and characterization of ARH3's MAR-hydrolase activity, we feel that generating ARH3, PARG and double knockdown cell lines and to measure basal levels of and stress-induced PARylation is out of the scope of this manuscript and would not provide additional insights for the mono-ADP-ribosylhydrolase activity of ARH3.

2. Page 11, the authors need to provide evidence supporting that trans-modified H3 is a MARYlated protein. Even though ARTD8 is known to be a mono-ART, the authors need to show what is the MARYlated residue (i.e. D/E, K, or S)? Then they need to use mass spectrometry to follow the deMARYlation of these residues, and demonstrate that ARH3 is a Ser-specific deMARYlating enzyme.

Response: We agree that it is important to show that ARH3 can clearly demodify targets that are solely MARYlated, which we do in Fig. 1b using automodified ARTD8 as target. This statement is additionally strengthened by the fact that PARG is not able to demodify ARTD8 (Fig. 1b). However, as to Fig. 1a (discussed on page 11), we do not claim that the ARTD1-mediated trans-modification of H3 is protein MARYlation. The observed demodification of H3 in Fig. 1a is most likely a combination of first demodification of PARylation and subsequently de-MARYlation of H3, since the demodification capacity of PARG is lower than the one of ARH3 (Fig. 1a). To clarify this aspect, the manuscript was revised accordingly (page 11).

Since we currently cannot exclude that ARH3 is also able to demodify other ADP-ribose acceptor sites than serines, we revised the text, abstract and the running title by removing the word “specific” and included a corresponding section to the discussion accordingly (page 1 and page 19).

3. Page 11, DraG needs to be consistent within the manuscript (all upper-case).

Response: DraG spelling was revised to be consistent throughout the manuscript.

4. Page 11, the sequence of the H2B peptide is wrong. The second amino acid (with ADP-ribosylation) should be Glu instead of Gln. Furthermore, the authors later claim that ARH3 specifically demodifies Ser-ADP-ribosylation. Why did they test ARH3 against a peptide that is ADP-ribosylated on Glu (or Gln)?

Response: The reviewer is right, that the amino acid sequence of H2B contains a Glu at position 2. However, since the chemical modification of Glu with ADP-ribose is currently not possible (i.e. it requires very low efficiency and highly complicated synthesis procedures), we replaced the Glu with Gln at position 2 of H2B to obtain a good yield of the modified peptide. The peptide used in this study is the only pure ADP-ribosylated peptide that is currently available in sufficient amounts and that cannot be hydrolyzed by any currently known ADP-ribosylhydrolase. This aspect has been clarified in the revised manuscript (page 6 and 11/12). The pull-down experiments with the modified peptides were thus performed with the aim to provide experimental evidence that ARH3 binds to a mono-ADP-ribosylated peptide whereas PARG cannot (revised Fig. 1d). We clarified this aspect in the revised manuscript on page 12.

5. Page 12, ARH3 and PARG need to be tested in one single experiment to assess their relative affinity for H2B peptides. The authors also need to elaborate on how they decouple the binding activity from the hydrolyzing activity? Once ARH3 binds to an ADP-ribosylated peptide, it is expected that ARH3 will rapidly hydrolyze its ADP-ribosylation moiety. This will complicate the interpretation of the resulting data.

Response: As requested by the reviewer we have repeated the experiments analysing ARH3 and PARG in the same experiments (revised Fig.1d).

With respect to the issue ‘binding vs. hydrolysis’, as explained above, the described Gln-ADPr linkage is artificial and not hydrolysable (we added this piece of information to the manuscript, page 6 and 11/12). Moreover, since the pulldowns were carried out on ice and ARH3 is not active at 4 °C (revised Sup. Fig. 1f) we can definitively exclude any hydrolase activity during the binding assays (page 13).

6. Page 13, line 391-394, the discussion here is confusing. They need to be re-worded

Response: We apologize for the unclear wording. The paragraph has been revised and is hopefully clearer now (page 13). The link to H₂O₂ is now discussed in the Discussion (page 18).

7. Page 13, line 410-415. the way Figure 2C is illustrated is very confusing. How did the author extract the data for PARG vs ARH3 treated samples, and how the data were cross-referenced? What is each point compared to? what are the S0 values? Also, the authors need to list the distribution of MARYlation on each amino acid.

Response: To increase the accuracy of the acceptor site identification, we compared and combined for this analysis the peptide (ADP-ribose acceptor site) identifications of this HCD experiment with our published ADP-ribose acceptor sites that were identified with high accuracy and confidence by EThcD MS (Bilan et al., Anal Chem, 2017) (Fig. 2c). To address which peptides and ADP-ribose acceptor sites are specifically de-modified by ARH3, we compared the ARH3-treated samples versus the PARG-treated samples and displayed only ADP-ribosylated peptides and their unmodified counterparts and their abundance by a volcano plot (Fig. 2c). This analysis revealed that ARH3 demodified most of the Af1521-enriched PARYlated/MARYlated peptides, while only a small set of MARYlated peptides could not be demodified by ARH3, and that these latter peptides are modified at arginine (page 13).

The value "S0" is a minimal fold change, which means even if a peptide provides a very good p-value in case the fold change is below that value (S0) it won't be significant. We revised the figure legend of Fig. 2c accordingly (page 21).

The requested list with the distribution of the amino acids functioning as acceptor sites was already included in the original manuscript (see Suppl Table 1).

8. Page 15, the authors claim that ARH3 demodifies Ser-ADPr residues in a KS motif. Why did they then use a GSK peptide, instead of GKS peptide when modeling the binding between this ADP-ribosylated peptide and ARH3 active site. Can the authors mutate the K in a substrate and show that this abolishes its binding to ARH3?

Response: We thank the reviewer for this very valid point. We repeated the modeling using a KSG peptide to place the modification in the middle of the peptide and to be consistent with the sequence obtained in Fig. 3e. The revision provided the same result as with the initial peptide (i.e. GSK), suggesting that our initial observation and discussion about the functionality of the K residue for the demodification are still valid. Fig. 3f was replaced with a revised figure panel and the text as well as the figure legend were revised accordingly (page 15).

We recently published that the KS motive is also found when analyzing the H₂O₂-induced ADP-ribosylome (Bilan et al., Anal Chem, 2017), suggesting that the K is required for the modification of proteins (data not shown). We mutated the K's of the H3 tail and modified both WT and mutant H3 peptides by ARTD1 and PARG to generate MARYlated substrates for ARH3 binding. These modification experiments revealed however that the K is important for the modification of the tail to be added, not allowing us currently to generate a modified substrate that lacks a K beside the S (data not shown).

Experimental proof for the role of K in the KS-ADP-ribosylation motive would require the synthesis of chemically modified peptides and binding assays, which is currently for technical reasons (instability of the modification) very difficult. We thus consider this to be out of the scope of this study.

Reviewer 2

I have carefully read the manuscript titled "Proteomic analyses identify ARH3 as a serine mono-ADP-ribosylhydrolase". Although the area of research is significant and the results interesting, I do not feel the results are of wide enough impact, nor the methods innovative enough to justify publication in Nature Communications. I would suggest a more archival journal for this work.

The main methods of the paper are described as mutagenesis, modeling (spelled modelling in abstract and modeling in methods, suggesting different authors wrote separate sections) and a mass spec technique to show in vivo function, both basal and oxidative stress initiated. Numerous mutants were identified that had an impact on activity - a pedestrian result. Not sure what modeling was used for. The methods described modeling as protein sequence alignment, protonation and MM minimization (CHARMM). This is not innovative and it is not even modeling. And the mass spec technique was reported in ref 42, pointing out the lack of innovation here as well.

Response: We agree with this reviewer that the applied techniques and methods themselves are not new, however, their use for this particular approach, for the question and project are novel and unique. The modeling consisted in structurally overlaying the C α atoms of an apo human ARH3 crystal structure with those of R. rubrum DraG (the latter containing an ADP-ribose ligand) with the graphical package WITNOTP. Following energy minimization of the ADP-ribosyl ligand in the ARH3 binding site, along with water molecules and protein atoms within 5 Å of the ligand, this ligand was step-wise modified (by addition of further atoms covalently linked to ADP ribose) and likewise energy minimized to obtain ADP-ribosylated serine, and ADP-ribosylated serine in the context of tripeptides acetyl-KSG and acetyl-RSG. The electrostatic potential was calculated in the continuum dielectric approximation by solving the finite-difference Poisson-Boltzmann equation with the Adaptive Poisson-Boltzmann Solver (APBS) software originating from Nathan Baker and colleagues, as per the cited reference. These molecular mechanics methods are established. They served to identify a possible binding interaction that helped guide mutational studies and that is consistent with the reported experimental results. Their application to ARH3 is new. Importantly, molecular modeling is not the main focus of our study, and we do not claim in our paper that novel modeling methods and/or protocols were used.

We revised also Fig. 3a since we realized during the revision of the manuscript that Fig. 3a was mislabeled, however, the conclusions of the paper remain valid.

Reviewer #1 (Remarks to the Author):

Abplanalp et al., submitted a revised manuscript describing the identification of ARH3 as a Ser mono-ADP-ribosylhydrolase. Although the authors have addressed a number of concerns and criticisms with augments and new experimental data, many of the results herein have been reported in a recent manuscript (Fontana et al., eLife, 2017). Importantly, the eLife paper was able to provide further evidence showing the amino acid specificity of ARH3, a key question the authors failed to address in their study. The authors also need to describe exactly how they "match" (line 426) the ADP-ribosylated peptides identified in this study to those from their previous dataset, and why this analysis is able to increase the accuracy and confidence in their ADP-ribosylation site assignment? Finally, the authors need to carefully proof-read their manuscript because it still contains many errors and typos (e.g., Y-axis label of Figure 1a and 1b).

NCOMMS-17-13941A

Point-to-point reply to reviewer's comments on our revision

Reviewer1:

“Abplanalp et al., submitted a revised manuscript describing the identification of ARH3 as a Ser mono-ADP-ribosylhydrolase. Although the authors have addressed a number of concerns and criticisms with augments and new experimental data, many of the results herein have been reported in a recent manuscript (Fontana et al., eLife, 2017). Importantly, the eLife paper was able to provide further evidence showing the amino acid specificity of ARH3, a key question the authors failed to address in their study.

Response: We have already included the indicated study in our last revised version. The experiments by Fontana et al. addressing the specificity of ARH3 were performed *in vitro* with single modified proteins and not at the proteomic wide level as in our study. Nevertheless, we have revised our discussion pointing to the fact that in the study by Fontana et al. only serine was found to be demodified by ARH3 (page 11).

The authors also need to describe exactly how they "match" (line 426) the ADP-ribosylated peptides identified in this study to those from their previous dataset, and why this analysis is able to increase the accuracy and confidence in their ADP-ribosylation site assignment?

Response: We describe more extensively the procedure to match the ADP-ribosylated peptides identified in this study to those from our previous dataset (see page 7 and Materials and Methods).

Finally, the authors need to carefully proof-read their manuscript because it still contains many errors and typos (e.g., Y-axis label of Figure 1a and 1b).”

Response: We carefully checked the manuscript again for potential errors and typos.